# Filtration rates of the manila clam, *Ruditapes philippinarum*, in tidal flats with different hydrographic regimes

**Bon Joo Koo**[1,2]*, **Jaehwan Seo**[1,2]

**1** School of Ocean Science, University of Science and Technology, Daejeon, Korea, **2** Marine Ecosystem Research Center, KIOST, Busan, Korea

* bjkoo@kiost.ac.kr

**Data Availability Statement:** All relevant data are within the manuscript and its Supporting Information files.

## Abstract

The manila clam *Ruditapes philippinarum* is widely distributed in the sandy mud sediments of tidal flats and plays a role in seawater purification by filtering suspended organic matter. This study was designed to evaluate differences in seawater purification based on the filtration rate of the manila clam in terms of particulate organic matter (POM) between two tidal flats with different hydrographic regimes. *In situ* experiments were carried out at Geunso and Sihwa tidal flats, along the west coast of Korea. The amount of POM removed from the water column by the feeding activity of the clam was measured in the field using a closed circulation chamber. The filtration rate of clams for POM at Sihwa tidal flat (2.86 for POC, 2.29 for PON and 5.46 L h$^{-1}$ gDW$^{-1}$ for Chl *a*) was higher than that at Geunso tidal flat (0.61 for POC, 0.89 for PON and 2.54 L h$^{-1}$ gDW$^{-1}$ for Chl *a*) which resulted from differences in the hydrographic regime, including tide characteristics, current speed and submergence time, and food quantity and quality. The current speed was much greater at Geunso tidal flat than at Sihwa tidal flat, but the submergence time by tide was longer at the latter site than the former, resulting in different feeding times for clams. The food quantity in terms of chlorophyll *a* was higher at Sihwa tidal flat than at Geunso tidal flat, and the food quality based on the C/N ratio of POM was better at the former site than the latter, with values of 12.8 and 15.6, respectively. These findings suggest that hydrographic regime could be important in understanding *in situ* filtration rates of *R. philippinarum*.

## Introduction

The manila clam, *Ruditapes philippinarum*, is widely distributed in the sandy mud sediments of tidal flats and shallows, with a geographic range including the west coasts of Korea and Japan, China, the northwestern USA, and European Atlantic coasts [1,2]. This clam is one of the most important shellfish in the Korean fisheries industry due to its high nutritional value and function of water purification in tidal flats [3]. In 2009, a total of 40,393 metric tons (MT) of clams were produced in Korea through aquaculture and fisheries, constituting the second greatest shellfish production, after oysters [4].

**Funding:** This research was supported by the Basic Science Research Program through the National Research Foundation of Korea (NRF) funded by the Ministry of Education (No. 2018R1D1A1B07049632, KIOST PN67930) and also was supported by a grant from the 'Development and security of mud shellfish resources in the small-sized tidal flat waterways', funded by the Ministry of Oceans and Fisheries, Korea. The funders had no role in study design, data collection and analysis, decision to publish, or preparation of the manuscript.

**Competing interests:** The authors have declared that no competing interests exist.

*R. philippinarum* is a filter-feeding bivalve responsible for the transport of materials at the sediment-water interface. It plays a role in seawater purification by filtering suspended organic materials from seawater, storing the filtered materials as assimilation products, and depositing excess organic matter onto the sediment as pseudofeces after ingestion [5]. The pseudofeces deposited onto sediments act as an important food source for deposit feeders. Bivalves such as *R. philippinarum* are characterized by relatively long lifespans and high biomass [6,7], which enables them to function as an organic matter reservoir in the material cycle of the tidal flat. In addition, the presence of an individual bivalve enhances biodiversity, as a space is formed between the particles of the matrix, providing habitat for various marine organisms.

Organic matter introduced into the marine ecosystem supports primary productivity in the water column, and enters the food chain through consumers; thus, it serves as a major link in the ecosystem's food chain, and is therefore a significant factor in maintaining ecosystem productivity. However, the high organic matter content of a contaminated water column is not transferred up through the food chain, but instead flows immediately into the sediment, causing a hypoxic layer in the water column and producing toxic substances such as sulfides in the sediments. Such sediments are the main culprit in deterioration of the environment near the seafloor. Filter feeders such as *R. philippinarum* block the capacity of organic matter to act as a contamination source by consuming organic matter from the water column as a food source [8].

Sihwa lake is an artificial lake formed by the construction of the Sihwa dike as part of the Great Reclamation Comprehensive Development Project, and is a representative area that underwent major environmental changes. Sihwa dike caused dissipation of intertidal zone with severe water quality deterioration. Thus, the government withdrew desalination and started the seawater exchange through sluice gate operation, as a result, tidal flat was partially created with repeated submergence and emergence. On the Sihwa tidal flat, introduction of organic matter directly into the sediment can cause pollution of the bottom-water environment, but the presence of filter feeders can prevent pollution by organic matter, as they decrease the amount of organic matter flowing into the sediment and are themselves consumed by higher-level consumers. In other words, the presence of filter feeders in Sihwa tidal flat can reduce the pollution load in the water column due to their use of organic matter that flows into the sediment as a food source, transferring it into the food chain. Therefore, the seawater purification capacity of filter feeders is essential as a method that can mitigate nutrient pollution.

Previous studies have reported effects of environmental factors such as temperature, salinity, particle size, and prey type on the filtration rate of this clam. The filtration of the clam increased with temperature up to 25˚C, with maximum rates in the range of 15–25˚C, and also increased with salinity up to 35 psu [3,9]. The filtration rate of this species for phytoplankton was much higher than that for bacteria, mainly due to differences in the size and biomass of these prey items [10]. In addition, the ingestion rate of the clam was much higher for small particles than for large particles due to the increased quantity of organic matter per unit volume [11]. However, scant research has been conducted on how the filtration rate of this species differs among hydrographic regimes.

Thus, the purpose of the present study is to evaluate differences in the seawater purification effect, based on the filtration rate of particulate organic matter (POM) by *R. philippinarum*, between two tidal flats with different hydrographic regimes.

## Study area

*In situ* experiments were carried out in two tidal flats along the west coast of Korea, Geunso (36˚44′7.43″N, 126˚10′40.12″E) and Sihwa (37˚17′10.01″N, 126˚40′57.54″E), with different

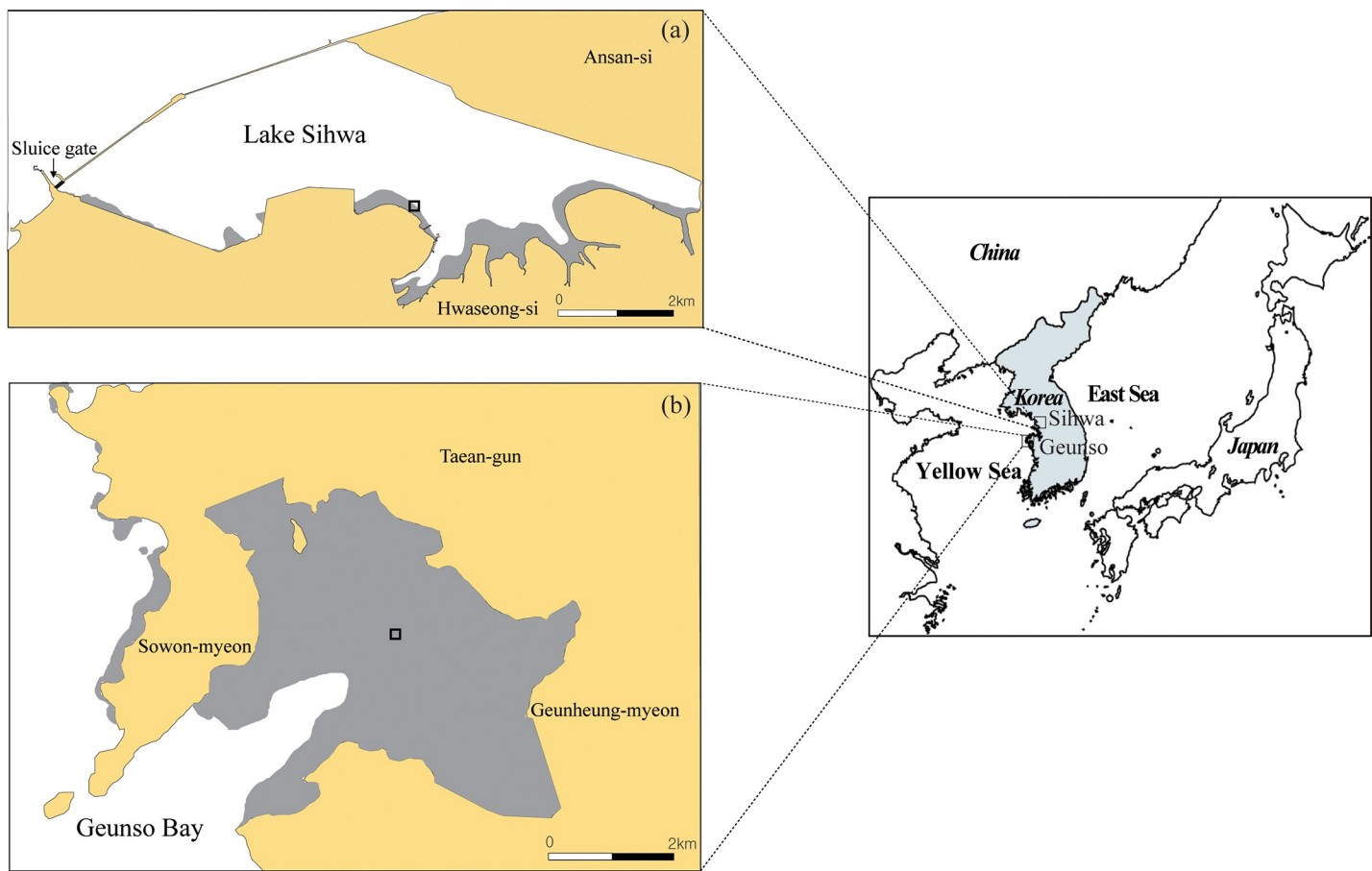

**Fig 1.** Locations and layout of the study sites: (A) Sihwa tidal flat and (B) Geunso tidal flat.

hydrographic regimes (Fig 1). A permission to conduct research in the two tidal flats was issued by the Ministry of Land, Transport and Maritime Affairs.

The manila clam, *Ruditapes philippinarum*, is widely distributed on the lower zone of these flats. Geunso tidal flat has a semidiurnal macro-tidal regime with a mean tidal range of 600 cm [12]. In the main tidal channel during spring tides, flood-current velocity ranges between 80 and 90 cm s$^{-1}$, and ebb-current velocity between 60 and 70 cm s$^{-1}$ [13]. While Geunso tidal flat is characterized by a macro-tidal regime, Sihwa tidal flat has an artificially controlled tide with a maximum range of 112 cm (Table 1).

**Table 1. Comparison of hydrographic regimes between Geunso and Sihwa tidal flats.**

|  | Geunso tidal flat | Sihwa tidal flat |
|---|---|---|
| Hydrographic characteristics | Macro-intertidal | Artificially controlled tide |
| Tidal range | 600 cm (mean) [a] | 112 cm (max) [b] |
| Current speed | ~ 90 cm s$^{-1}$ [c] | 0.1 ~ 33.1 cm s$^{-1}$ [d] |

[a][12]

[b][14]

[c][13]

[d][15]

Sihwa tidal flat is located along Lake Sihwa, which is brackish (27 psu). Due to the limited water exchange between the inner lake and outer sea, which occurs only through a narrow sluice gate in accordance with the tidal cycle (see Fig 1), the tidal range in Lake Sihwa is smaller, and its speed slower (33.1 cm s$^{-1}$ maximum), compared to Geunso. The experimental sites on these flats also have differing environmental characteristics. The surface sediment type in Sihwa tidal flat is sandy, with a mean grain size of 3.1 to 3.3 Φ, while Geunso tidal flat has sandy silt of 4.2 to 4.8 Φ.

## Materials and methods

### 1) *In situ* experiments

**(1) Experimental equipment.** The amount of POM removed from the water column through the feeding activity of *R. philippinarum* was measured in the field using a closed circulation chamber (Fig 2).

The chamber was constructed from transparent acrylic, with an inner diameter of 0.49 m, height of 0.6 m, cross-sectional area of 0.19 m$^2$, and volume of 113 L. The bottom of the

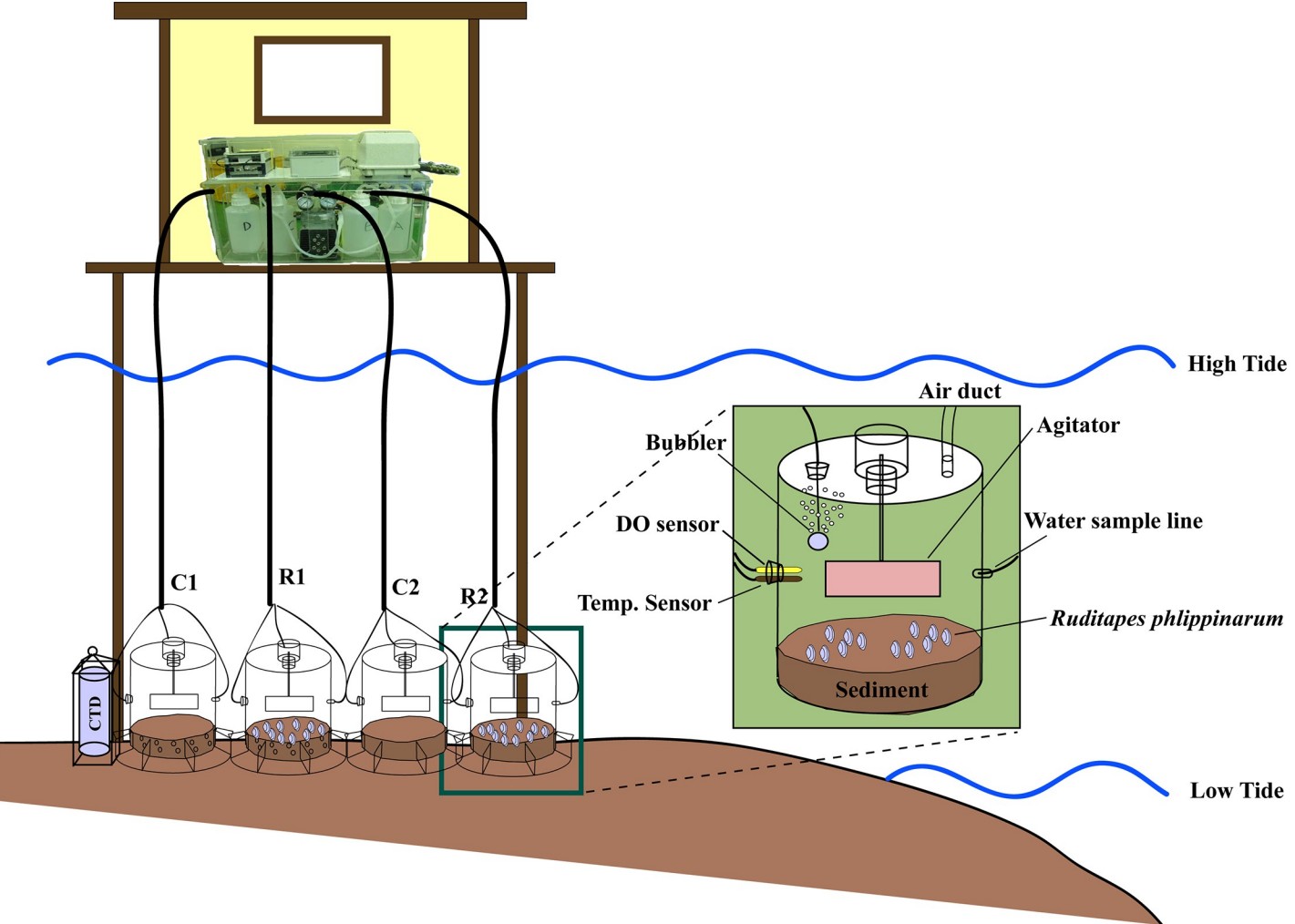

**Fig 2. Schematic diagram of the in situ experiment to measure filtration rates of *Ruditapes philippinarum*.** C1 and C2 represent control chambers without clams and R1 and R2 represent experimental chambers containing clams.

chamber was closed and the top was equipped with a lid that could be opened and closed. An agitator was installed inside the chamber, connected to a plastic tube through a hole in the lid, which was intended to prevent natural settling of suspended solids (SS) and stagnation of seawater in the chamber during the experiment, and to maintain homogeneity of the medium. The connection to the lid was waterproofed so that seawater could not flow into the chamber, even when it was flooded. A vacuum pump was connected to the opposite end of a plastic pipe in turn connected to the inside of the chamber, so that seawater in the chamber could be collected from the outside. Due to pressure generated inside the chamber during water sampling, an air pipe connecting the chamber with the atmosphere was installed to equalize the pressure. Dissolved oxygen was monitored to prevent a decrease in metabolic activity due to depletion of dissolved oxygen in the chamber, and when the concentration decreased below 80%, oxygen was supplied using an aerator installed inside the chamber. Because the field experiment was conducted during a submergence period, all devices connected to the chamber were installed on a tower, where an observer was located (Fig 2).

**(2) Installation of chambers and sampling.** Experiments were carried out in the habitat of *R. philippinarum* in Geunso and Sihwa tidal flats in October 2009 in the same manner (Fig 3).

Four chambers were installed in the sediments, buried to a depth of 10 cm, and connected to the support structure of the tower with an iron frame to prevent movement due to tidal currents. Inside the chamber, previously dried and prepared defaunated sediments were installed at a depth of 10 cm. Defaunated sediments were prepared by sieving field sediments to remove macroinvertebrates, including *Ruditapes*, and then dried at a constant temperature of 27˚C for 10 days. After the dried sediment was placed in the chamber, the upper part of the chamber was sealed using a plastic sieve with 0.3-mm mesh and acclimated in seawater in the field for 3 days. During the acclimation period, 2- and 3-year-old clams were collected from the field and immediately transplanted into the chamber, and then acclimated for 1 day with the sediments. Fifty individuals were transplanted into each of two chambers (*Ruditapes* chambers) and none were transplanted into the other two chambers (control chambers). Shortly before starting the experiment, the lid of the chamber was closed and the peristaltic pump, air duct, Fibox-3 oxygen meter, water temperature gauge, agitator, and aerator were installed. Seawater from the experimental site was allowed to flow into the chamber during the flood tide. At this time, to prevent the sediment in the chamber from being disturbed by the influent seawater, silicone plugs installed at various heights in the chamber wall were opened, beginning with the plug in the lowest position. After seawater had filled the chamber, all plugs were closed and the agitator was employed to maintain a constant seawater flow rate in the chamber of 5 cm s$^{-1}$. We filled a chamber in the laboratory with the same sediment and seawater as the chamber installed at the experimental site prior to the experiment, and determined the seawater flow rate (5 cm s$^{-1}$) from the agitator at which organic matter in the sediment was not suspended and natural sedimentation of POM was prevented. Based on the flow rate obtained under laboratory conditions, the flow rate of seawater in the chamber was maintained during the experimental period.

After 10 min of agitator operation, seawater samples were collected from the four chambers using a vacuum pump. Water sample collection was performed four times (Sihwa tidal flat) or five times (Geunso tidal flat) at regular intervals until the experimental area was exposed during the ebb tide to determine the mean concentration of POM in the experimental shore region, a seawater sample was simultaneously collected at the same depth as the experimental chamber. At the time of collection of each seawater sample, 1 L of seawater was sampled from each chamber and from the surrounding water column. From these samples, 750 mL of seawater was placed into a polyethylene container to measure the concentrations of SS, particulate organic carbon (POC), and particulate organic nitrogen (PON), and the remaining 250 mL

(a)

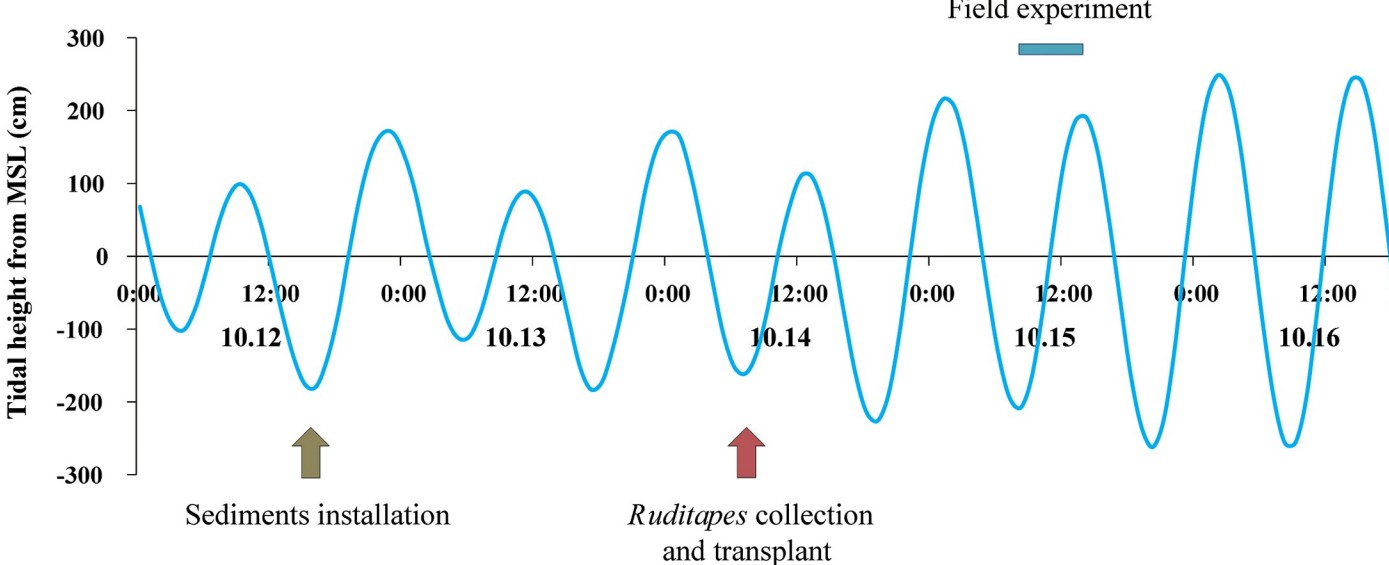

(b)

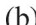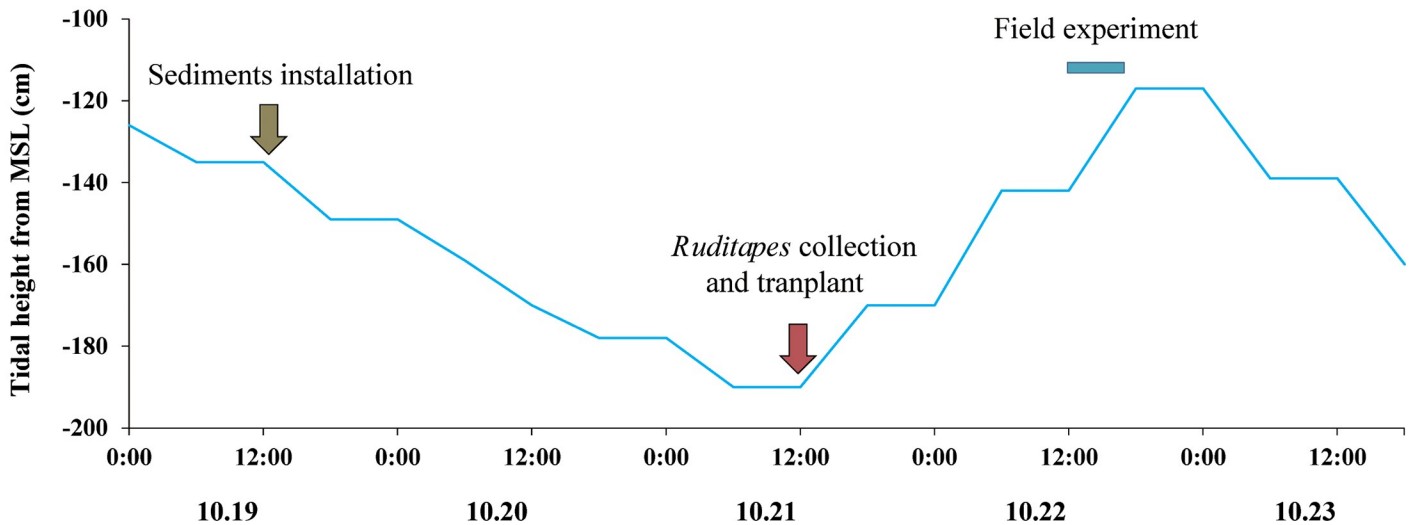

**Fig 3.** Diagrams showing the sequence of the in situ experiment to measure filtration rates of *Ruditapes philippinarum* scheduled according to the tide at (A) Sihwa tidal flat and (B) Geunso tidal flat.

was filtered using GF/F filter paper, then frozen and stored for chlorophyll *a* analysis. Chambers were equipped with aerators ("bubblers" in Fig 2) to prevent dissolved oxygen from dropping below 80% and inhibiting the metabolic activity of clams, but these were not necessary to activate during the experiment at either site. After the experiment, clams were captured for measurement of shell length, flesh wet weight, and flesh dry weight. During the experiment, the dissolved oxygen, and water temperature of the seawater were simultaneously measured at the same depth as the chamber (CTD SBE-19; Sea-Bird Electronics, USA).

## 2) Sample analysis

**(1) Total suspended solids.** First, 0.7-μm Whatman GF/F filter paper was washed three times with 20 mL of distilled water in a vacuum filter apparatus, dried at 103–105˚C for 1 hour, and weighed using an electronic scale. At the experimental site, seawater collected in plastic bottles (polypropylene [PP]; Nalgene, USA) was mixed thoroughly and 100 ml was filtered onto GF/F filter paper. The filter paper was washed three times with 10 ml distilled water for desalination. The remaining material on the filter paper was dried at 105–110˚C for 2 hours and then weighed. The SS content was determined from the difference between the wet and dry weights of the filter paper.

**(2) Particulate organic carbon and particulate organic nitrogen.** To determine the concentrations of POC and PON, 0.7-μm Whatman GF/F filter paper, through which 500 ml of seawater had been filtered (burned at 550˚C for 5 hours before use), was lyophilized; carbon and nitrogen were then removed by placing the filter in a desiccator containing hydrochloric acid solution for 24 hours. Aliquots were placed into tin cups after measuring their weight, and concentration analysis was performed using an elemental analyzer (EA1110; CE Instruments, UK).

**(3) Chlorophyll *a*.** Using 0.7-μm Whatman GF/F filter paper, 250 mL of the seawater sample was filtered, and the filter was transferred to the laboratory frozen. The filter paper was placed in a centrifuge tube with 10 mL of 90% acetone and then shaken thoroughly, and the filter paper was allowed to react in a dark, refrigerated room for 24 hours. The extracted chlorophyll *a* was measured using a spectrophotometer (Cary 50; Varian, USA).

## 3) Calculation of POM filtration rate in the water column

Assuming that the feeding activity of the clams is constant, POM in the chamber should be removed at a constant rate such that its concentration should decrease exponentially over time. In this experiment, chlorophyll *a*, POC and PON were used as indicators of POM. If the initial concentration of the indicator substance is C0 and the coefficient of the concentration decrease is Z, the concentration Ct after t time is defined by the following equation.

$$Ct = C0 \cdot e^{-zt}$$

The coefficient of the concentration decrease, Z, is defined as follows:

$$Z = -\ln (Ct/C0) \cdot t$$

The concentration decrease rate, d, per unit time of the indicators in water is expressed as follows:

$$d = 1 - e^{-z}$$

The filtration rate (FR) per unit time is given by the following equation, where V is the volume of the chamber and h is the unit of time.

$$FR \ (L/h) = V \cdot d = V \cdot (1 - e^{-z})$$

Because the amount of particulate matter in the chamber is limited, as the observation time becomes longer, the rate of concentration decrease becomes smaller, and thus the filtration rate can be underestimated. In this study, the filtration rate was calculated for the period in which the maximum filtration was observed from among the time periods. The filtration rate of POM was adjusted by natural sedimentation rate from control chamber.

A two-sample *t* test was used to determine differences between samples. The results were considered statistically significant when $p < 0.05$.

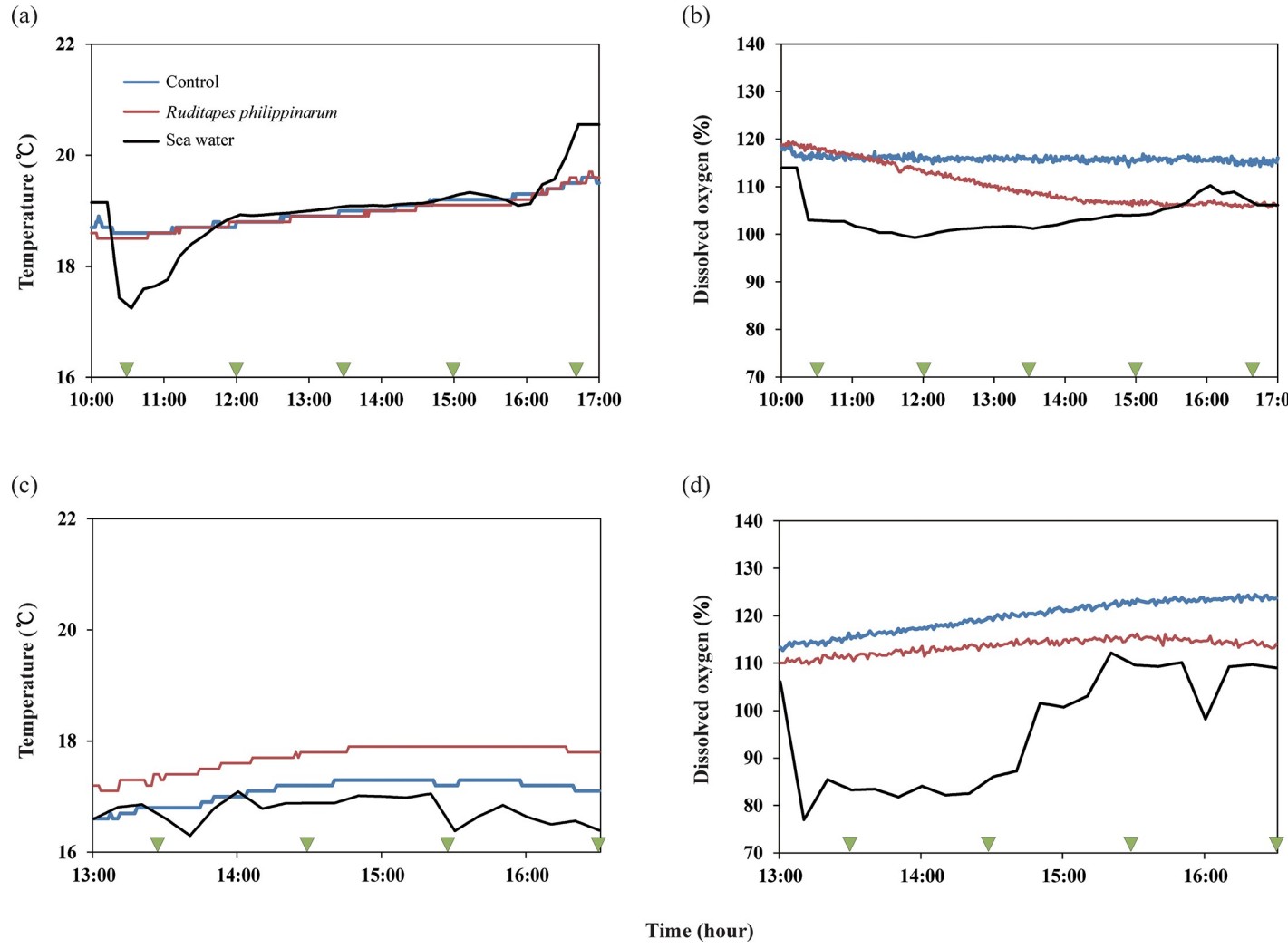

**Fig 4.** Variations in temperature and dissolved oxygen concentration in two experimental chambers and seawater over the entire study period at Geunso and Sihwa tidal flats: (A) temperature at Geunso tidal flat, (B) dissolved oxygen at Geunso tidal flat, (C) temperature at Sihwa tidal flat, (D) dissolved oxygen at Sihwa tidal flat. Green triangles represent sampling times.

## Results

### 1) Water temperature and dissolved oxygen

In Sihwa tidal flat, the water temperatures in the *Ruditapes* chamber, the control chamber, and the surrounding seawater did not change significantly during the experiment, and ranged from 16.3 to 17.3°C (Fig 4). The water temperature in the *Ruditapes* chamber was highest during experiment with mean values of 17.7±0.2°C, followed by control chamber (17.1±0.2°C) and seawater (16.8±0.2°C). Although, there was statistical significance ($p<0.05$), but difference of temperature was not differ greatly between the *Ruditapes* chamber and the control chamber; due to the closed environment, the water temperature inside the chamber was slightly higher than that of the seawater around the chamber ($p<0.05$).

In Geunso tidal flat, the seawater temperature showed greater changes than in Sihwa tidal flat. During the flood tide, as the current flowed at shallow depths, the water temperature

reached 19.1˚C, but as the water depth increased, the water temperature decreased to 17.2˚C. Thereafter, the temperature rose rapidly due to rising atmospheric temperature and tended to increase gradually from around the time of slack tide. During the ebb tide, as surface water that had been heated during the daytime passed through the experimental area, the water temperature increased to 21.2˚C (Fig 4). Unlike Sihwa tidal flat, the mean water temperature was not significantly different among the *Ruditapes* chamber, the control chamber, and the seawater with values of 18.9±0.3, 18.9±0.3 and 19.0±0.7˚C in Geunso tidal flat, respectively ($p < 0.05$). The mean water temperatures of *Ruditapes* chamber, the control chamber, and the seawater were significantly higher in Geunso tidal flat than in Sihwa tidal flat in entire experiment ($p < 0.05$) and chambers were less variable in water temperature compared to the seawater.

During the experiment, dissolved oxygen exceeded 100% in all chambers at both study sites. In Sihwa tidal flat, dissolved oxygen in the *Ruditapes* chamber and the control chamber increased continuously compared to their initial values, and the control chamber showed a larger increase. However, the dissolved oxygen concentration in the chamber showed a different tendency in the Geunso tidal flat. The control chamber maintained a constant concentration, while the *Ruditapes* chamber showed a decreasing trend. Changes in the dissolved oxygen concentration of seawater were greater in the Geunso tidal flat than Sihwa tidal flat. The dissolved oxygen in the seawater was generally lower than in the chambers during entire experiment in both tidal flats.

## 2) Suspended solids

The concentration of SS in the seawater of Sihwa tidal flat did not change much during the experiment. At the beginning of the experiment, the SS concentration was 0.10 g L$^{-1}$. After 1 hour, the SS concentration increased slightly to 0.12 g L$^{-1}$ and then decreased to 0.091 g L$^{-1}$ at the end of the experiment (Fig 5A).

The SS concentrations of the *Ruditapes* and control chambers were both about 0.10g L$^{-1}$, similar to that of the seawater at the study site. The control chamber did not show significant changes in SS concentration over time, while the *Ruditapes* chamber showed a pronounced decrease in SS concentration. In particular, there was a significant decrease in SS concentration (0.076 g L$^{-1}$) over a period of 1 hour, beginning after 1 hour had elapsed in the experiment.

Changes in SS concentrations at Geunso tidal flat differed significantly from those observed in Sihwa tidal flat. The SS concentrations in the two chambers, were initially identical at 0.14 g L$^{-1}$, higher than those observed in Sihwa tidal flat (Fig 6A).

The concentration in the seawater around the study site was highest at the beginning of the experiment, then decreased rapidly over time, and increased again at the end of the experiment. This pattern can be attributed to the fact that the flood current during the flood tide and the ebb current during the ebb tide caused the sediment to be resuspended. The SS concentrations in the *Ruditapes* and control chambers decreased rapidly until 1 hour and 30 minutes had elapsed. After this time point, the changes in SS concentrations in the two chambers showed differing tendencies. There was no significant change in SS concentration in the control chamber, while SS content decreased continuously in the *Ruditapes* chamber. The sharp decrease in SS concentration during the initial stage of the experiment was attributed to the seawater flow rate in the chamber (5 cm s$^{-1}$ in this experiment) being insufficient to prevent deposition of SS resuspended by the flood current. However, after a certain amount of SS settled, a constant SS concentration was maintained in the control chamber during the experimental period, while it continued to decrease in the *Ruditapes* chamber, leading to concentration differences between the two chambers, as observed at Sihwa tidal flat.

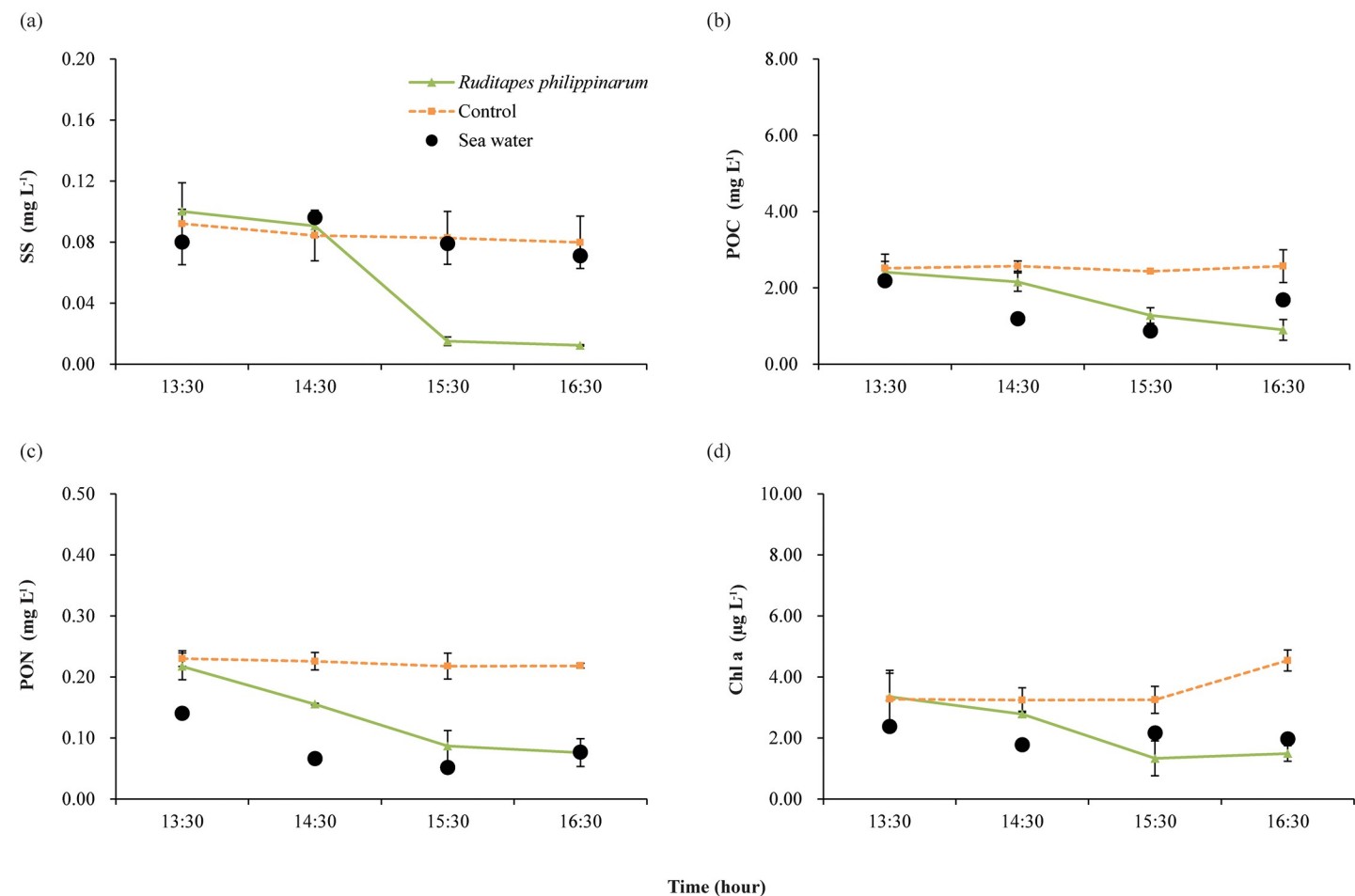

**Fig 5.** Variation in suspended solids (SS) and particulate organic matter (POM) in two experimental chambers and seawater over the entire study period at Sihwa tidal flat: (A) SS (B) particulate organic carbon (POC) (C) particulate organic nitrogen (PON) and (D) chlorophyll *a*. The error bars represent 95% confidence intervals.

### 3) POC and PON

Concentrations of POC and PON in the seawater ranged from 0.88 to 2.18 mg L$^{-1}$ and 0.05 to 0.14 mg L$^{-1}$, respectively, during the experiment period at Sihwa tidal flat (Fig 5B). The concentrations of these substances in the seawater tended to gradually decrease and then increase. The POC and PON concentrations showed similar tendencies in both chambers (Fig 5B and 5C). In the control chamber, there was no significant change in the concentrations of the two indicators during the experimental period, whereas there was a large decrease in these concentrations due to the filter-feeding activity of clams in the *Ruditapes* chamber. During the experimental period, the POC concentration decreased from an initial value of 2.40 to 0.89 mg L$^{-1}$, and the PON concentration decreased from an initial value of 0.22 to 0.07 mg L$^{-1}$. For both indicators, the greatest decrease in concentration occurred between the second and third sampling periods, when the concentration of POC decreased from 2.15 to 1.27 mg L$^{-1}$ and the concentration of PON decreased from 0.16 to 0.08 mg L$^{-1}$.

The initial concentrations of POC and PON at Geunso tidal flat were higher than those at Sihwa tidal flat. POC and PON concentrations in the seawater were in the range of 1.00 to 6.56 mg L$^{-1}$ and 0.05 to 0.43 mg L$^{-1}$, respectively (Fig 6B and 6C). In seawater at the study site, the concentration of each indicator tended to gradually decrease and then increase. In the chamber, there

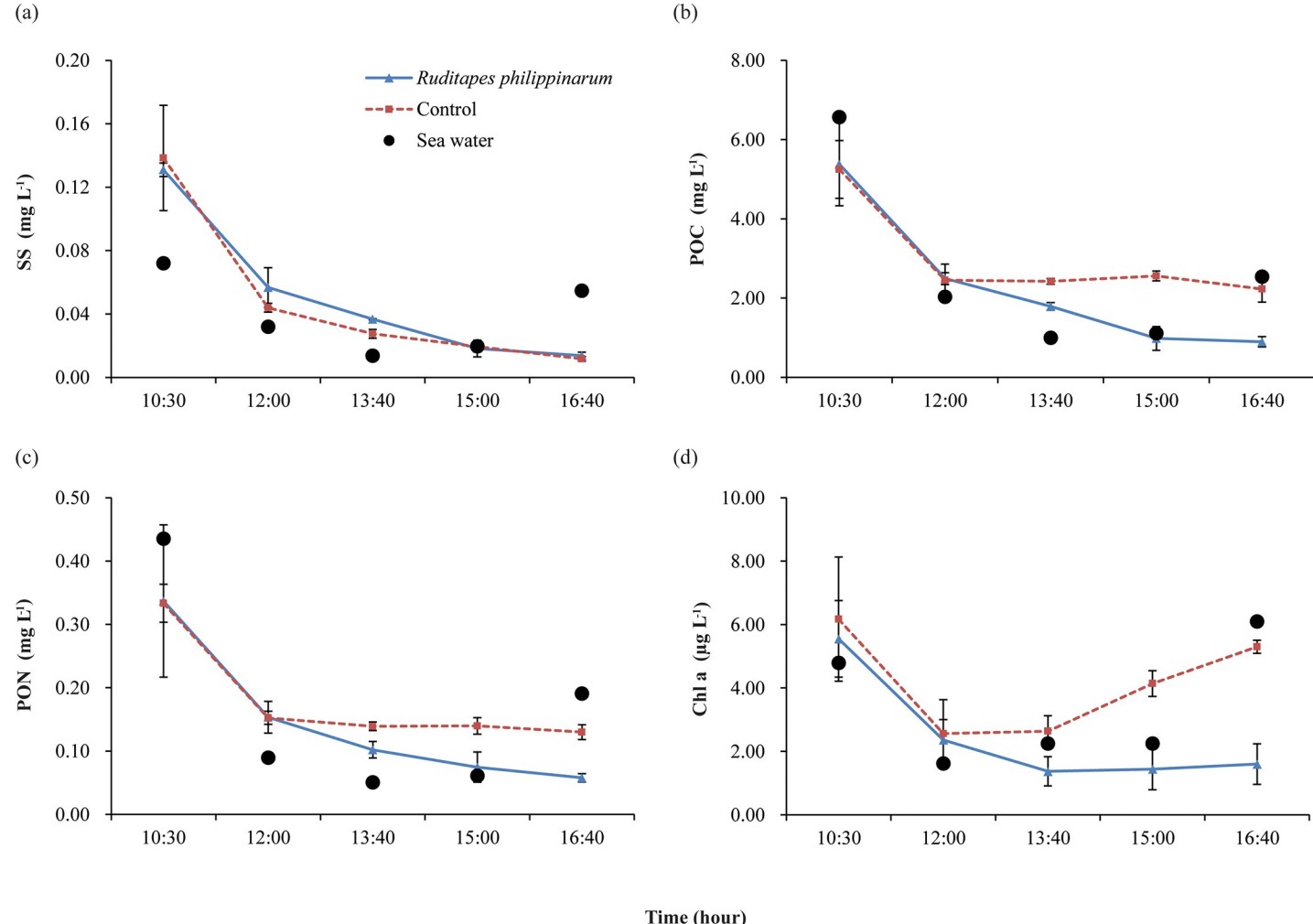

**Fig 6.** Variation in SS and POM in two experimental chambers and seawater over the entire study period at Geunso tidal flat: (A) SS (B) POC (C) PON and (D) chlorophyll *a*. The error bars represent 95% confidence intervals.

was a sharp decrease in these concentrations at the beginning of the experiment, as observed for suspended matter, and a difference in concentration between the *Ruditapes* chamber and the control chamber arose during the second sampling period. After the second sampling event, no significant change in the concentrations of the two indicators was observed in the control chamber, but their concentrations decreased sharply due to the filter-feeding activity of clams in the *Ruditapes* chamber. In the *Ruditapes* chamber, the POC concentration decreased from 2.49 to 0.89 mg L$^{-1}$, and the PON concentration decreased from 0.15 to 0.06 mg L$^{-1}$.

### 4) Chlorophyll *a*

The chlorophyll *a* concentration in seawater at the Sihwa tidal flat was 2.37 µg L$^{-1}$. This value was relatively high due to resuspension of sediment during the initial phase of the experiment, and a constant concentration of about 2.0 µg L$^{-1}$ was maintained thereafter (Fig 5D). At the beginning of the experiment, the concentration was about 3.3 µg L$^{-1}$ in both the control and *Ruditapes* chambers, but the difference in chlorophyll *a* concentration between the two chambers increased gradually over time. In the control chamber, little change in the chlorophyll *a* concentration occurred

during the first 2 hours (3.24 to 3.27 $\mu$g L$^{-1}$), but it then increased dramatically to 4.54 $\mu$g L$^{-1}$ at the end of the experiment. On the other hand, in the *Ruditapes* chamber, a large decrease in concentration was observed, down to 1.49 $\mu$g L$^{-1}$ at the end of the experiment.

At Geunso tidal flat, the chlorophyll *a* concentration was relatively high at the beginning and end of the experiment due to resuspension caused by the flood and ebb currents. The chlorophyll *a* concentration at the beginning and the end of the experiment was 4.79 and 6.10 $\mu$g L$^{-1}$, respectively (Fig 6D). The chlorophyll *a* concentration showed a sharply decreasing tendency during the initial part of the experiment, followed by a constant concentration around the time of the high-water stand, and then increased sharply during the ebb tide. In the *Ruditapes* and control chambers, a sharp decrease in the chlorophyll *a* concentration took place during the beginning of the experiment, as described for POC and PON concentrations. At the second sampling time, the chlorophyll *a* concentration was similar between the *Ruditapes* and control chambers, at 2.36 and 2.56 $\mu$g L$^{-1}$, respectively. The difference in chlorophyll *a* concentration between the two chambers at the third sampling time was 1.27 $\mu$g L$^{-1}$, with a lower concentration in the *Ruditapes* chamber. Thereafter, no significant changes in the chlorophyll *a* concentration in the *Ruditapes* chamber occurred, while that in the control chamber tended to increase (S1 Table).

### 5) The filtration rate of *R. philippinarum*

The morphometric dimensions of *R. philippinarum* were compared between Geunso and Sihwa tidal flats (Table 2). The shell length, height and width were significantly higher in Geunso tidal flat than in Sihwa tidal flat. The flesh wet weight and flesh dry weight were also higher in the former than in the latter with significance

The filtration rate of *R. philippinarum* was calculated from the changes in the concentrations of POC, PON, and chlorophyll *a* in the *Ruditapes* chamber. As noted above, because the supply of POM introduced with food source materials at the beginning of the experiment was limited in the closed chamber, the filtration rate may be underestimated due to the reduced concentration with increased observation time. Therefore, the filtration rate was calculated for the period in which the maximum filtration was observed from among the time periods. The filtration rate at Sihwa tidal flat was 2.86±1.28 L h$^{-1}$ gDW$^{-1}$ for POC, 2.29±1.85 L h$^{-1}$ gDW$^{-1}$ for PON, and 5.46±1.07 L h$^{-1}$ gDW$^{-1}$ for chlorophyll *a* (Table 3).

At Geunso tidal flat, the filtration rates were lower than at Sihwa tidal flat. The filtration rates of POC, PON and chlorophyll *a* were 0.61±0.04, 0.89±0.19, and 2.54±1.61 L h$^{-1}$ gDW$^{-1}$, respectively. Although the difference in POM filtration rate was not statistically significant, it was higher at Sihwa tidal flat than that at Geunso tidal flat.

## Discussion

In contrast to our expectation that dissolved oxygen would decrease due to the respiration of clams, no significant decrease was observed in any chamber, and it exceeded 100% throughout

**Table 2. Comparison of morphometric data of *Ruditapes philippinarum* between Geunso and Sihwa tidal flats.**

|  | Shell length (mm) | Shell height (mm) | Shell width (mm) | Flesh wet weight (g) | Flesh dry weight (g) |
|---|---|---|---|---|---|
| Geunso (n = 94) | 38.5±3.1 | 27.3±2.4 | 19.2±2.1 | 6.2±1.6 | 0.4±0.1 |
| Sihwa (n = 93) | 33.2±2.3 | 22.7±1.6 | 15.3±1.2 | 2.2±0.6 | 0.2±0.1 |
| P value | <0.05 | <0.05 | <0.05 | <0.05 | <0.05 |

Significant differences by *t* test at 0.05

**Table 3. Comparison of filtration rate of *Ruditapes philippinarum* between Geunso and Sihwa tidal flats.**

| | Filtration rate (L h-1 gDW-1) | | |
|---|---|---|---|
| | **POC** | **PON** | **Chl *a*** |
| Geunso | 0.61±0.04 | 0.89±0.19 | 3.17±1.72 |
| Sihwa | 2.86±1.28 | 2.29±1.85 | 5.46±1.07 |
| P value | 0.24 | 0.40 | 0.17 |

POC, particulate organic carbon; PON, particulate organic nitrogen; Chl *a*, chlorophyll *a*; Significant differences by *t* test at 0.05

the experiment. Moreover, oxygen concentration was elevated in the *Ruditapes* and control chambers placed at Sihwa, presumably because the depth of the chambers was shallow, which allowed photosynthesis by phytoplankton. This result was attributed to the oxygen consumption by the clams being offset by the oxygen production due to photosynthesis in the chamber.

Generally, filtration rate in terms of filtering ability for seawater increase as clam size decrease. This relationship may be due to the decrease with age of the gill surface to body size ratio and to the higher metabolic demands of the younger individuals [16,17]. Segade et al., 2003; Sylvester et al., 2005; Han et al., 2008). Han et al. (2008) reported that the filtration rate of small *Ruditapes philippinarum* (0.2 gDW) was about 28% higher than that of large ones (0.4 gDW) at 20°C. This indicated that the filtration rate of *Ruditapes* can be changed by size and biomass, but, the difference in filtration rate is too great to be explained by size and biomass in this study.

One experimental condition that must be considered carefully when calculating the amount of POM removed from seawater by filter feeders using a closed chamber, as indicated in this study, is that the concentration of POM introduced into the chamber at the beginning of the experiment should be maintained. While the concentrations of the indicators in the control chamber at Sihwa tidal flat were constant during the experimental period, the concentrations of those indicators (POC, PON and Chlorophyll *a*) decreased rapidly in the control chamber at Geunso tidal flat (Fig 6). This result may be due to settling of some of the POM initially introduced into the chamber in the sediments, which is related to the intensity of the flood current in the tidal flats. At Sihwa tidal flat, which has a closed circulation environment, the inflow of resuspended particulate matter from the sediment into the chamber was limited because the intensity of the flood current was low. In contrast, the intensity of the flood current was high at Geunso tidal flat, and resuspended sediments were introduced into the chamber in large quantities. Prior to this experiment, the flow rate in the chamber that prevented resuspension of particulate matter without causing sedimentation of POM from seawater introduced into the chamber was determined under laboratory conditions using sediments and seawater from Sihwa tidal flat (5 cm s$^{-1}$). This flow rate was applied to the experiments at both sites, and was slower than in the *in situ* environment at Geunso tidal flat. However, the concentration of particulate matter in the control chamber remained constant after a certain period of time, whereas that in the *Ruditapes* chamber tended to decrease; thus, it is possible to obtain a filtration rate based on this difference in concentration. However, it is possible that the filtration rate at Geunso tidal flat measured in this experiment was somewhat underestimated, as we presume that filtration by clams also occurred during the period when the concentration decreased rapidly at the beginning of the experiment.

At Sihwa tidal flat, there is no natural tidal rhythm, and ebb and flood currents are created by artificial manipulation of the sluice gate. These ebb and flood currents are not created twice

a day like natural currents, but are instead repeated at an interval of 2 or 3 days. The mean daily submergence time of the study site at Sihwa tidal flat was longer than that at Geunso tidal flat, with values of 21 and 16 h day$^{-1}$, respectively. Therefore, clams at the former site can carry out feeding activity more efficiently, and for a longer time, than those at the latter site.

The relatively high filtration rate at Sihwa tidal flat can be attributed to this difference in hydrographic regimes. Another factor affecting the difference in filtration rate is high productivity in the water column at Sihwa tidal flat, which causes a quantitative difference in the food available to Manila clams. The Manila clams at Sihwa tidal flat had a higher growth rate than those at Geunso tidal flat, and the mean chlorophyll *a* concentration in the water column was significantly higher at the former site than the latter, with values of 3.1 and 52.5 μg L$^{-1}$, respectively [18]. This result suggests that the quantity of food available contributed to differences in the filtration rate of this clam between the two tidal flats.

The C/N ratio of POM is an indicator of food quality. High values of the C/N ratio are generally indicative of nutrient starvation, with lower values being recorded under non-limiting nutrient conditions [19]. Optimal feeding by *Ruditapes* occurred at C/N ratios of 8.4–10.5, with a range of 6.2–12.0, and was correlated with nitrogen rather than carbon content [20]. The mean C/N ratios of the Geunso and Sihwa tidal flats over the entire study period were 15.6 and 12.8, respectively, and the mean PON concentration at Sihwa tidal flat was slightly higher than that at Geunso tidal flat, with values of 0.13 and 0.10 mg L$^{-1}$, respectively. These results suggest that the food quality at Sihwa is better than that at Geunso tidal flat, resulting in a difference in filtration rate between these two tidal flats.

The filtration rates obtained in this study, especially at Sihwa tidal flat, were higher than those reported in previous studies. While Lee [21] in similar field experimental settings as the current study reported the filtration rate of the Manila clams ranging from 0.13 to 0.97 L h$^{-1}$ gDW$^{-1}$, our study reports significantly higher values (2.29 to 5.46 L h$^{-1}$ gDW$^{-1}$), which could be related to differences in experiment methods. In their experiment, the flow rate in the chamber was maintained at 65 cm s$^{-1}$, which was 13 times higher than that used in the present study. At flow rates of 15–20 cm s$^{-1}$ or higher, the bottom sediments and feces of bivalves begin to be resuspended [22]. Maintaining a flow rate of 65 cm s$^{-1}$ in the chamber caused resuspension of sediments, meaning that POM was continuously supplied from the sediments to the water column during the experiment. Thus, the decrease in POM concentration due to clams was offset by the supply from sediments, and the filtration rate of clams was consequently underestimated.

In another field experiment, Aoyama and Suzuki [23] reported the filtration rate of 3.44 L h$^{-1}$ gDW$^{-1}$. They conducted their experiment by adding 0.6–1.0 L of diatoms (a mixture of *Chaetoceros* sp. and *Skeletonema* sp.) cultured in a laboratory into the culture solution to greatly increase the initial concentrations of indicators in the chamber. These high initial concentrations promoted feeding activity of clams and a relatively high filtration rate was observed. In other words, these findings indirectly suggest that the feeding activity of clams may increase in areas where a large amount of food is available in the seawater, as was the case at Sihwa tidal flat. This likely explains our finding that the filtration rate at Sihwa tidal flat was higher than that at Geunso tidal flat.

In conclusion, this study evaluated differences in the filtration rate of *R. philippinarum* due to differing hydrographic regimes between two tidal flats that the influence of differing hydrographic conditions on the filtration rates of Manila clams has not been studied previously. Though this study was limited by factors such as salinity, turbidity and particle size that affect variations in filtration rates, the findings provide important information about the effects of hydrographic regime on the filtration rate of this species.

## Supporting information

**S1 Table. Variations on mean concentration of SS and POM in two experimental chambers and seawater over the entire study period at Geunso and Sihwa tidal flats.**
(DOCX)

**S2 Table. Comparison of filtration rate of *Ruditapes philippinarum* for POM between Geunso and Sihwa tidal flats over time.**
(DOCX)

## Acknowledgments

We would like to thank Dr. Jennifer Ruesink and anonymous reviewer for their valuable comments.

## Author Contributions

**Formal analysis:** Bon Joo Koo, Jaehwan Seo.

**Investigation:** Bon Joo Koo, Jaehwan Seo.

**Writing – original draft:** Bon Joo Koo.

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
