## [Decision Letter · Decision Letter 0]

6 Aug 2019

PONE-D-19-15055

Filtration rates of the manila clam, Ruditapes philippinarum, in tidal flats with different hydrographic regimes

PLOS ONE

Dear Dr Koo,

Thank you for submitting your manuscript to PLOS ONE. After careful consideration, we feel that it has merit but does not fully meet PLOS ONE’s publication criteria as it currently stands. Therefore, we invite you to submit a revised version of the manuscript that addresses the points raised during the review process.

ACADEMIC EDITOR: Please respond critically to each of the issues raised, specifically, the statistical treatment of the data. Also, limit your the data you report to the ones you collected, especially in the abstract so that not to mislead readers. Consistency is important in scientific writing; please make sure that your results and discussion are consistent and the flow is maintained. 

We would appreciate receiving your revised manuscript by Sep 20 2019 11:59PM. To enhance the reproducibility of your results, we recommend that if applicable you deposit your laboratory protocols in protocols.io, where a protocol can be assigned its own identifier (DOI) such that it can be cited independently in the future. For instructions see: http://journals.plos.org/plosone/s/submission-guidelines#loc-laboratory-protocols

We look forward to receiving your revised manuscript.

Kind regards,

Ismael Aaron Kimirei, Ph.D.

Academic Editor

PLOS ONE

Journal Requirements:

2, In your Methods section, please provide additional information regarding the permits you obtained for the work. Please ensure you have included the full name of the authority that approved the field site access and, if no permits were required, a brief statement explaining why." 2) please send the following request and do not ping with follow-up: "In your Methods section, please provide additional location information, including geographic coordinates for the data set if available.

Reviewers' comments:

Reviewer's Responses to Questions

**Comments to the Author**

1. Is the manuscript technically sound, and do the data support the conclusions?

Reviewer #1: Partly

Reviewer #2: Yes

2. Has the statistical analysis been performed appropriately and rigorously? 

Reviewer #1: No

Reviewer #2: Yes

3. Have the authors made all data underlying the findings in their manuscript fully available?

Reviewer #1: No

Reviewer #2: Yes

4. Is the manuscript presented in an intelligible fashion and written in standard English?

Reviewer #1: Yes

Reviewer #2: No

5. Review Comments to the Author

Reviewer #1: This manuscript provides data from a 4-hour study of water properties in 63-L in situ containers on two tidal flats, comparing how water properties change with and without manila clams. The overall intent involves some clever field equipment to get measurements of what clams are doing in situ. On the other hand, the statistical methods are currently insufficient (and insufficiently-documented) to reveal if a) clams affect water properties, and b) this effect differs between the two sites. (And c) if this effect differs over time during the 4-hour study) Since there are just two replicates of each chamber type, the power to detect a difference is low, but such a statistical approach is essential to scientific assessment of hypotheses. More details on this issue are provided in the line-by-line comments below.

Abstract line 29-30: inconsistent results – stated to be higher chl at Sihwa with 3.1 ug/L, vs. Geunso at 52.5 ug/L – clearly Geunso is higher than Sihwa by the numbers. 52.5 ug/L chl at Geunso is not evident from any results in the paper, for instance, Fig. 6 shows chlorophyll concentrations to be on the order of 5 ug/L. I think it is inappropriate to use 52.5 and 3.1 ug/L in the abstract, as these data were not collected in the present study but are cited from other work.

Line 56: ecosystem homeostasis? Many ecologists – I guess myself included – would take issue with a sense of ecosystems self-regulating to avoid change. The construction of the sentence also implies that organic matter (“it”) contributes to homeostasis. The authors should consider whether they really mean ecosystem productivity supported by organic matter, or ecosystem resilience to nutrient inputs achieved through suspension-feeders?

Line 62: Sihwa tidal flat is first mentioned here in the paper, without any description of where in the world it is, or why its function is worthy of note. I recommend a description here that introduces Sihwa, its size and construction, and its management challenges. This will motivate why clams could serve to mitigate water pollution there.

Line 69: alternative to what? Seems like it’s possible to instead say “as a method that can mitigate nutrient pollution or impaired water quality.”

Line 93 “riparian” refers to the out-of-water habitat adjacent to a water body, usually a river. Thus a tidal flat cannot be a riparian area, since it is covered with water (and also not a river): drop “is a riparian area that”

Line 102: what is the frequency of high and low tides at the Sihwa site? Based on Fig. 3, tidal period is a major feature that distinguishes the two sites!

Line 150: authors need to provide information on the biomass of clams in each chamber. It is stated (line 175) that clams were sacrificed at the end of the experiment for biomass and size measurements; these results are important for readers to know, because filtration rates can change with bivalve size, or high biomasses of clams may undergo exploitation competition that reduces apparent filtration rate.

Line 157 and 164: I think I understand that the seawater flow rate was not a continuous flow through the container, but rather the speed at which water moved around within the container – was this more like a current, or more like turbulence? And from later information I assume that removing 5x 1L from the 63-L chambers did not require adding any new water? So the water sampled in the chamber at the beginning of the submergence period was the same as at the end?

Line 165: more important to state the time between successive samples than how many samples were collected, given that ultimately filtration rate is based on the steepest decline between successive samples, not fitting all data

Line 175: Oxygen did not drop below 80% so the aerator was not used? Or because the aerator acted to elevate oxygen concentrations? I think from the figures it would be accurate to say: “Chambers were equipped with aerators (“bubblers” in Fig. 2) to prevent dissolved oxygen from dropping below 80% and inhibiting the metabolic activity of clams, but these were not necessary to activate during the experiment at either site.”

Line 176-178: very sparse description of the calculations used to calculate removal rate by comparing clam and no-clam containers, and how to distinguish filtration from sedimentation. Do you mean “Removal rate of suspended material from the water by clams was evaluated by the change in water properties in the clam-containing chambers over time, adjusted by changes that were occurring concurrently in the control chambers. Specifically, the change between successive samples in each control chamber was subtracted from the change in the paired clam chamber, because losses from the water column in the control chamber were due to natural sedimentation rate.”

Line 179-180: not clear how measuring water column particles before and after experiment in all containers would be a metric of natural sedimentation

Line 197: does the desiccator contain HCl? Or does the desiccator contain beakers of HCl, one for each sample?

Line 171 says 250 ml were used for chl analysis, but line 202 says 300 ml

Lines 208-225 only address calculations for filtration rate, whereas the authors also need to consider how they are going to test statistically whether the patterns in the two treatments and two tidal flats are different. Asserting these differences by examining time series or by comparing the magnitude of two means, as is currently the case in the results, is not sufficient support for the claims and conclusions.

Line 225: maximum slope should be specified to refer to Z (logarithmic change in concentration), rather than the absolute difference in concentration. Also, were these calculations carried out only with the clam-containing containers, or based on the difference from reference containers? And how were the two replicates of each treatment handled? i.e. calculate the average concentration at each time based on two chambers before doing the calculations, or do the calculations for each chamber separately? And if the latter, and if including how the water properties in these chambers change relative to controls, then how were the chambers with and without clams paired up?

Line 231: specify “surrounding seawater”

Line 230-253: Multiple statements exist in these paragraphs about significance, or one group being different from another. Typically such statements in science need to be accompanied by a statistical test. So this section needs some rewriting (or the authors need to set up the statistical tests – I think these are less important here for the physical conditions of the samples, and more important to have statistical tests associated with comparing reference and control chambers and the two sites). Also, some of the major distinctions are not mentioned, for instance: a) thoughout 4 hours, temperature with clams at Geunso is 0.5 degrees warmer than without clams, b) dissolved oxygen outside the chambers is generally below – sometimes a lot below – the concentrations inside the chambers, and c) chambers seem to be less variable in water temperature than the water they are bathed in.

In the discussion section, the authors provide possible explanations for differing filtration rates between the two tidal flats, and why these may differ from other studies. This general content is relevant and important. Several comments here regarding logical connections drawn in the discussion. 1) I find it puzzling that the higher quality of food at Sihwa relative to Geunso can be used as an explanation for higher filtration rate at Sihwa, because in the present study, the food concentrations and quality appear very similar (although no statistical test has been done). The authors need to be careful about using results from other studies (about growth rates and chlorophyll concentrations) to provide a mechanism in the current case, since I think that the filtration rate response for this mechanism is expected to be immediate. 2) The other mechanism that the authors provide for the higher filtration rate at Sihwa is in regards to circatidal rhythms, but it appears that these might also be immediate responses – that is, ebb and flood tides reduce filtration not because of inherent circatidal rhythms but because rapid water motion resuspend sediment that interferes with feeding. 3) Overall in the discussion, the authors attribute differences in filtration to hydrographic regime, but need to keep in mind that the filtration rate measurements were done in closed chambers that had identical water motion conditions, so the clams would have to respond to longer-term factors than what they are experiencing during the experiment; yet the authors look at the changes over time in filtration rate as a response to immediate conditions.

Fig. 5, 6 description – “two experimental chambers” is confusing. Does this refer to replicates or treatments?

Fig. 7 description – insufficient to have the only material about statistical tests here. Statistical tests need to be set up in the methods, with results in tables or supplemental material. The authors need to clarify what is a “replicate” and whether the comparisons indicated by letters a and be are only done within sites or also across sites. I am afraid that the authors are considering “replicates” as filtration rate detected by chl, PON, and POC. Rather, a replicate needs to be a chamber.

Table 2 as currently written is redundant with the text of results.

In all figures, authors need to define what is shown by error bars – I assume SD of two replicates, but have really no idea!

Data availability: authors should provide raw data for measurements of clams and water properties. The data provided in figures are based on a series of calculations.

Reviewer #2: The purpose of this study was to investigate the filtration rates of the Manila clam between two natural sites. The test device is widely suggested to use for the study on the physiology of the benthic shellfish. Because we always test the filtration rate or ammonia excretion rate in the lab. It might be just 1 or 2 factors for the experiment. This paper show us a interesting device to test the filtration rate at the complicate environments, including the temperature, salinity, suspended particles, current speed, tidal flat, and chlorophyll a. The results provide important information for the role of the Manila clam in the ecology and aquaculture. But the grammars of the manuscript should be carefully revised before publication. Some suggestions are as below,

Line 19: manila clam should be Manila clam, the same changes should be made on line 89.

Lines 28 and 30: give more details for the latter and former;

Line 51: In addition, the…. What is purpose to show this sentence at this paragraph?

Line 78-80 this part should be rewritten to show more significant purposes for this study.

Line 174-175 How can you measure the dissolved oxygen, shell length and body weight?

Line 175 body length should be shell length.

Line 229-253 provide more environmental factors in this part, such as salinity, ammonia, nitrogen and phosphate, because these factors are also effected the filtration rate on Manila clam.

Line 452-455 this part should be rewritten to conclude the main results for this manuscript, and explain how to use the findings to do the further study.

The English of the discussion should be polished by the English speaker, and more references should be added to support the findings of this study.

The figures are obscure. The clear figures should be provided for publication.

6. PLOS authors have the option to publish the peer review history of their article (what does this mean?). If published, this will include your full peer review and any attached files.

Reviewer #1: Yes: Jennifer Ruesink

Reviewer #2: No

---

## [Author Response · Author response to Decision Letter 0]

25 Oct 2019

5. Review Comments to the Author

Reviewer #1: This manuscript provides data from a 4-hour study of water properties in 63-L in situ containers on two tidal flats, comparing how water properties change with and without manila clams. The overall intent involves some clever field equipment to get measurements of what clams are doing in situ. On the other hand, the statistical methods are currently insufficient (and insufficiently-documented) to reveal if a) clams affect water properties, and b) this effect differs between the two sites. (And c) if this effect differs over time during the 4-hour study) Since there are just two replicates of each chamber type, the power to detect a difference is low, but such a statistical approach is essential to scientific assessment of hypotheses. More details on this issue are provided in the line-by-line comments below.

Abstract line 29-30: inconsistent results – stated to be higher chl at Sihwa with 3.1 ug/L, vs. Geunso at 52.5 ug/L – clearly Geunso is higher than Sihwa by the numbers. 52.5 ug/L chl at Geunso is not evident from any results in the paper, for instance, Fig. 6 shows chlorophyll concentrations to be on the order of 5 ug/L. I think it is inappropriate to use 52.5 and 3.1 ug/L in the abstract, as these data were not collected in the present study but are cited from other work.

→ These values were deleted in abstract (L 35-36).

Line 56: ecosystem homeostasis? Many ecologists – I guess myself included – would take issue with a sense of ecosystems self-regulating to avoid change. The construction of the sentence also implies that organic matter (“it”) contributes to homeostasis. The authors should consider whether they really mean ecosystem productivity supported by organic matter, or ecosystem resilience to nutrient inputs achieved through suspension-feeders?

→ We intended to mean ecosystem productivity, thus, it was corrected to ecosystem productivity (L 62-63).

Line 62: Sihwa tidal flat is first mentioned here in the paper, without any description of where in the world it is, or why its function is worthy of note. I recommend a description here that introduces Sihwa, its size and construction, and its management challenges. This will motivate why clams could serve to mitigate water pollution there.

→ The description of Sihwa tidal flat was added in L 69-74: Sihwa lake is an artificial lake formed by the construction of the Sihwa dike as part of the Great Reclamation Comprehensive Development Project, and is a representative area that underwent major environmental changes. Sihwa dike caused dissipation of intertidal zone with severe water quality deterioration. Thus, the government withdrew desalination and started the seawater circulation through sluice gate operation, as a result, tidal flat were partially created with repeated submergence and emergence.

Line 69: alternative to what? Seems like it’s possible to instead say “as a method that can mitigate nutrient pollution or impaired water quality.”

→ It was corrected according to your suggestion as follows: Therefore, the seawater purification capacity of filter feeders is essential as a method that can mitigate nutrient pollution (L 75-77).

Line 93 “riparian” refers to the out-of-water habitat adjacent to a water body, usually a river. Thus a tidal flat cannot be a riparian area, since it is covered with water (and also not a river): drop “is a riparian area that”

→ It was corrected according to your suggestion.

Line 102: what is the frequency of high and low tides at the Sihwa site? Based on Fig. 3, tidal period is a major feature that distinguishes the two sites!

→ Due to the Sihwa tidal flat has an artificially controlled tide, it is different from tidal cycle of general tidal flat. The mean frequency of high and low tides in Sihwa tidal flat was 7 times and 10 times in the month when we conducted the experiment, respectively.

Line 150: authors need to provide information on the biomass of clams in each chamber. It is stated (line 175) that clams were sacrificed at the end of the experiment for biomass and size measurements; these results are important for readers to know, because filtration rates can change with bivalve size, or high biomasses of clams may undergo exploitation competition that reduces apparent filtration rate.

→ We added the morphometric information of clams in Table 3. Generally, filtration rate in terms of filtering ability for seawater increase as clam size decrease. This relationship may be due to the decrease with age of the gill surface to body size ratio and to the higher metabolic demands of the younger individuals (Segade et al., 2003; Sylvester et al., 2005; Han et al., 2008). Han et al. (2008) reported that the filtration rate of small Ruditapes philippinarum (0.2 g DW) was about 28% higher than that of large one (0.4 g DW) at 20 °C. This indicated that the filtration rate of Ruditapes can be changed by size and biomass, but, the difference on filtration rate is too great to explain by size and biomass in this study. The results and discussion were described in the each section.

Line 157 and 164: I think I understand that the seawater flow rate was not a continuous flow through the container, but rather the speed at which water moved around within the container – was this more like a current, or more like turbulence? And from later information I assume that removing 5x 1L from the 63-L chambers did not require adding any new water? So the water sampled in the chamber at the beginning of the submergence period was the same as at the end?

→ The seawater flow rate was not a continuous flow but it was more like a current. The volume of each chamber is 113 L, we couldn’t find 63 L chamber volume as you mentioned. But, there was no addition any new water during the experiment.

Line 165: more important to state the time between successive samples than how many samples were collected, given that ultimately filtration rate is based on the steepest decline between successive samples, not fitting all data

→ The water samples were collected at regular intervals with an hour for Sihwa tidal flat and an hour and forty mins for Geunso tidal flat, respectively. It is shown in Figs. 5 and 6 at x-axis.

Line 175: Oxygen did not drop below 80% so the aerator was not used? Or because the aerator acted to elevate oxygen concentrations? I think from the figures it would be accurate to say: “Chambers were equipped with aerators (“bubblers” in Fig. 2) to prevent dissolved oxygen from dropping below 80% and inhibiting the metabolic activity of clams, but these were not necessary to activate during the experiment at either site.”

→ Oxygen concentration in the chamber did not drop below 80% during experiment period in both sites. Thus, aerator was not used as you mentioned. The part of oxygen concentration was corrected according to your suggestion as follows: Chambers were equipped with aerators (“bubblers” in Fig. 2) to prevent dissolved oxygen from dropping below 80% and inhibiting the metabolic activity of clams, but these were not necessary to activate during the experiment at either site.

Line 176-178: very sparse description of the calculations used to calculate removal rate by comparing clam and no-clam containers, and how to distinguish filtration from sedimentation. Do you mean “Removal rate of suspended material from the water by clams was evaluated by the change in water properties in the clam-containing chambers over time, adjusted by changes that were occurring concurrently in the control chambers. Specifically, the change between successive samples in each control chamber was subtracted from the change in the paired clam chamber, because losses from the water column in the control chamber were due to natural sedimentation rate.”

→ We did not measure natural sedimentation, thus, it was deleted.

Line 179-180: not clear how measuring water column particles before and after experiment in all containers would be a metric of natural sedimentation

→ We did not measure natural sedimentation, thus, it was deleted.

Line 197: does the desiccator contain HCl? Or does the desiccator contain beakers of HCl, one for each sample?

→ The desiccator contained beakers of HCL one for each sample.

Line 171 says 250 ml were used for chl analysis, but line 202 says 300 ml

→ It was corrected to 250 mL in L 212.

Lines 208-225 only address calculations for filtration rate, whereas the authors also need to consider how they are going to test statistically whether the patterns in the two treatments and two tidal flats are different. Asserting these differences by examining time series or by comparing the magnitude of two means, as is currently the case in the results, is not sufficient support for the claims and conclusions.

→ The aim of this study is to evaluate difference on filtration rate between two tidal flats with different hydrographic regimes. Because the amount of particulate matter in the chamber is limited, as the observation time becomes longer, the rate of concentration decrease becomes smaller, and thus the filtration rate can be underestimated as we described. Thus, we think that time series filtration rate can’t reflect actual decreasing tendency of that.

Line 225: maximum slope should be specified to refer to Z (logarithmic change in concentration), rather than the absolute difference in concentration. Also, were these calculations carried out only with the clam-containing containers, or based on the difference from reference containers? And how were the two replicates of each treatment handled? i.e. calculate the average concentration at each time based on two chambers before doing the calculations, or do the calculations for each chamber separately? And if the latter, and if including how the water properties in these chambers change relative to controls, then how were the chambers with and without clams paired up?

→ The calculations of POM filtration rate in the water column were carried out only with the clam-containing chambers and were calculated based on the average concentration at which maximum slope of the decreasing concentration in two chambers.

Line 231: specify “surrounding seawater”

→ It was corrected to surrounding seawater in L 241.

Line 230-253: Multiple statements exist in these paragraphs about significance, or one group being different from another. Typically such statements in science need to be accompanied by a statistical test. So this section needs some rewriting (or the authors need to set up the statistical tests – I think these are less important here for the physical conditions of the samples, and more important to have statistical tests associated with comparing reference and control chambers and the two sites). Also, some of the major distinctions are not mentioned, for instance: a) thoughout 4 hours, temperature with clams at Geunso is 0.5 degrees warmer than without clams, b) dissolved oxygen outside the chambers is generally below – sometimes a lot below – the concentrations inside the chambers, and c) chambers seem to be less variable in water temperature than the water they are bathed in.

→ We conducted statistical analysis in water temperature between chambers and seawater as well as both tidal flats and additional results were described in this section as follows : (a) The water temperature in the Ruditapes chamber was highest during experiment with mean values of 17.7±0.2 °C, followed by control chamber (17.1±0.2 °C) and seawater (16.8±0.2 °C). Although, there was statistical significance (p<0.05), but difference of temperature was not differ greatly between between the Ruditapes chamber and the control chamber; due to the closed environment, the water temperature inside the chamber was slightly higher than that of the seawater around the chamber (p<0.05). (b) Unlike Sihwa tidal flat, the mean water temperature was not significantly different among the Ruditapes chamber, the control chamber, and the seawater with values of 18.9±0.3, 18.9±0.3 and 19.0±0.7 °C in Geunso tidal flat, respectively (p<0.05). The mean water temperatures of Ruditapes chamber, the control chamber, and the seawater were significantly higher in Geunso tidal flat than in Sihwa tidal flat in entire experiment (p<0.05) and chambers were less variable in water temperature compared to the seawater. (c) The dissolved oxygen in the seawater was generally lower than in the chambers during entire experiment in both tidal flats.

In the discussion section, the authors provide possible explanations for differing filtration rates between the two tidal flats, and why these may differ from other studies. This general content is relevant and important. Several comments here regarding logical connections drawn in the discussion. 1) I find it puzzling that the higher quality of food at Sihwa relative to Geunso can be used as an explanation for higher filtration rate at Sihwa, because in the present study, the food concentrations and quality appear very similar (although no statistical test has been done). The authors need to be careful about using results from other studies (about growth rates and chlorophyll concentrations) to provide a mechanism in the current case, since I think that the filtration rate response for this mechanism is expected to be immediate. 2) The other mechanism that the authors provide for the higher filtration rate at Sihwa is in regards to circatidal rhythms, but it appears that these might also be immediate responses – that is, ebb and flood tides reduce filtration not because of inherent circatidal rhythms but because rapid water motion resuspend sediment that interferes with feeding. 3) Overall in the discussion, the authors attribute differences in filtration to hydrographic regime, but need to keep in mind that the filtration rate measurements were done in closed chambers that had identical water motion conditions, so the clams would have to respond to longer-term factors than what they are experiencing during the experiment; yet the authors look at the changes over time in filtration rate as a response to immediate conditions.

→ 1) Although, the initial chl a concentration of this study was similar in both tidal flats, however, the annual mean value of that was significantly higher in Sihwa tidal flat than in Geunso tidal flat. This suggests that the differences of food quantity and quality likely caused different feeding activity which resulted in different filtration rate between both tidal flats. 

2) We filled a chamber in the laboratory with the same sediment and seawater as the chamber installed at the experimental site prior to the experiment, and determined the seawater flow rate (5 cm s-1) from the agitator at which organic matter in the sediment was not suspended and natural sedimentation of POM was prevented. Thus, we think that the feeding activity of clams was unaffected by that. Kim et al. (1999) have reported that activity rhythms of clams are controlled not only by exogenous factors, but also by an endogenous circatidal periodicity. They showed that manila clams removed from their natural environment and maintained for 9 weeks in continuously immersed conditions exhibited a clear endogenous circatidal rhythm. Therefore, we think that the different filtration rate between two tidal flats was caused by inherent circatidal rhythms not immediate responses.

3) The aim of this study is to evaluate difference on filtration rate between two tidal flats with different hydrographic regimes. Because the amount of particulate matter in the chamber is limited, as the observation time becomes longer, the rate of concentration decrease becomes smaller, and thus the filtration rate can be underestimated as we described. Thus, we think that time series filtration rate can’t reflect actual decreasing tendency of that.

Fig. 5, 6 description – “two experimental chambers” is confusing. Does this refer to replicates or treatments?

→ Two experimental chambers represent replicates.

Fig. 7 description – insufficient to have the only material about statistical tests here. Statistical tests need to be set up in the methods, with results in tables or supplemental material. The authors need to clarify what is a “replicate” and whether the comparisons indicated by letters a and be are only done within sites or also across sites. I am afraid that the authors are considering “replicates” as filtration rate detected by chl, PON, and POC. Rather, a replicate needs to be a chamber.

→ The letters a and b represent comparison of mean filtration rate within sites. The mean filtration rate represents POC, PON and chl a removal rate, thus, the error bars indicate 95% confidence intervals of each filtration rate.

Table 2 as currently written is redundant with the text of results.

In all figures, authors need to define what is shown by error bars – I assume SD of two replicates, but have really no idea!

→ Table 2 (Table 3 in revised manuscript) is a main result in this experiment which can help see it to readers. The error bars in figures were defined.

Data availability: authors should provide raw data for measurements of clams and water properties. The data provided in figures are based on a series of calculations.

→ We added morphometric information of clams in Table 2 and water property data was provided in supplement.

 

Reviewer #2: The purpose of this study was to investigate the filtration rates of the Manila clam between two natural sites. The test device is widely suggested to use for the study on the physiology of the benthic shellfish. Because we always test the filtration rate or ammonia excretion rate in the lab. It might be just 1 or 2 factors for the experiment. This paper show us a interesting device to test the filtration rate at the complicate environments, including the temperature, salinity, suspended particles, current speed, tidal flat, and chlorophyll a. The results provide important information for the role of the Manila clam in the ecology and aquaculture. But the grammars of the manuscript should be carefully revised before publication. Some suggestions are as below,

Line 19: manila clam should be Manila clam, the same changes should be made on line 89.

→ It was corrected.

Lines 28 and 30: give more details for the latter and former;

→ We couldn’t understand what you mean exactly.

Line 51: In addition, the…. What is purpose to show this sentence at this paragraph?

→ We intended to describe the important function of bivalve in the ecosystem.

Line 78-80 this part should be rewritten to show more significant purposes for this study.

→ We think that this sentence implies the purpose of this study well.

Line 174-175 How can you measure the dissolved oxygen, shell length and body weight?

→ The dissolved oxygen was measured by Fibox-3 oxygen meter. It was described in material section (L 165). The shell length was measured with calipers and the body weight was obtained by weighing.

Line 175 body length should be shell length.

→ It was corrected.

Line 229-253 provide more environmental factors in this part, such as salinity, ammonia, nitrogen and phosphate, because these factors are also effected the filtration rate on Manila clam.

→ We did not measure additional environmental factors such as salinity, ammonia, nitrogen and phosphate.

Line 452-455 this part should be rewritten to conclude the main results for this manuscript, and explain how to use the findings to do the further study.

→ It was corrected as follows: This study is significant because it evaluated differences in the filtration rate of R. philippinarum due to differing hydrographic regimes between two tidal flats that has not been studied. Though this study was limited by factors that affect variations in filtration rate, the findings provide important information about the effects of hydrographic regime on the filtration rate of this species.

The English of the discussion should be polished by the English speaker, and more references should be added to support the findings of this study.

→ The English in this document has been checked by at least two professional editors, both native speakers of English. For a certificate, please see: http://www.textcheck.com/certificate/ASji1i We added more references.

The figures are obscure. The clear figures should be provided for publication.

 → We have uploaded clear figures but we have no idea why that were obscured in PDF file. We think that it is likely due to technical problem.

---

## [Decision Letter · Decision Letter 1]

26 Nov 2019

PONE-D-19-15055R1

Filtration rates of the manila clam, Ruditapes philippinarum, in tidal flats with different hydrographic regimes

PLOS ONE

Dear Dr Koo,

Thank you for submitting your manuscript to PLOS ONE. After careful consideration, we feel that it has merit but does not fully meet PLOS ONE’s publication criteria as it currently stands. Therefore, we invite you to submit a revised version of the manuscript that addresses the points raised during the review process.

ACADEMIC EDITOR: pleased  respond to all the comments by reviewer #1 on the validity of your treatments and the statistics. The comments are explicit and understandable. It should be possible to follow them and make necessary corrections or revisions to your MS and submit for further consideration or provide a rebuttal accordingly. I would consider this MS one more time.

We would appreciate receiving your revised manuscript by Jan 10 2020 11:59PM. To enhance the reproducibility of your results, we recommend that if applicable you deposit your laboratory protocols in protocols.io, where a protocol can be assigned its own identifier (DOI) such that it can be cited independently in the future. For instructions see: http://journals.plos.org/plosone/s/submission-guidelines#loc-laboratory-protocols

We look forward to receiving your revised manuscript.

Kind regards,

Ismael Aaron Kimirei, Ph.D.

Academic Editor

PLOS ONE

Reviewers' comments:

Reviewer's Responses to Questions

**Comments to the Author**

1. If the authors have adequately addressed your comments raised in a previous round of review and you feel that this manuscript is now acceptable for publication, you may indicate that here to bypass the “Comments to the Author” section, enter your conflict of interest statement in the “Confidential to Editor” section, and submit your "Accept" recommendation.

Reviewer #1: (No Response)

Reviewer #2: All comments have been addressed

2. Is the manuscript technically sound, and do the data support the conclusions?

Reviewer #1: No

Reviewer #2: Yes

3. Has the statistical analysis been performed appropriately and rigorously? 

Reviewer #1: No

Reviewer #2: I Don't Know

4. Have the authors made all data underlying the findings in their manuscript fully available?

Reviewer #1: No

Reviewer #2: Yes

5. Is the manuscript presented in an intelligible fashion and written in standard English?

Reviewer #1: Yes

Reviewer #2: Yes

6. Review Comments to the Author

Reviewer #1: As I wrote before, it is not appropriate to combine measurements of three different components of the seston (POC, PON, chla), all collected in the same chamber at the same time, to calculate a mean filtration rate. Mean filtration rate could be based on the filtration of the same component of the water across the two experimental chambers at each site, after each is adjusted for the change in the companion reference chamber. A t-test could then be done using two samples at each site, since there were two experimental chambers (containing clams) at each site. (Note: I am actually uncertain whether the “four chambers” on line 155 refer to four chambers per tidal flat, or four chambers in total. If the latter, then there is not a possibility to compare the two tidal flats statistically, as there is only one filtration value for each flat. But in their response the authors write “two experimental chambers represent replicates” so I think they should be able to do statistics with n=2 at each tidal flat.)

Line 230: How were the control chambers used in the calculations of filtration rate? The authors write: “the filtration rate was calculated for the period in which the maximum slope of the decreasing indicator concentration was observed.” There are three issues to address: 1) The maximum slope at Sihwa (Fig 5) occurs in the first time interval, but is equivalent for clam and reference chambers, such that no net filtration would be indicated IF the values were properly adjusted by comparison with reference; 2) the maximum slope seems to indicate the steepest portion of Figures 5 and 6, rather than the portion with the largest exponent in the equation for filtration; 3) selecting a time of maximum filtration from among the time periods creates potential bias if clam filtration at one site is more variable than at the other. The authors should do the following: 1) determine which reference chamber to pair with each clam chamber, 2) adjust Ct in the clam chamber so that it accounts for change in the reference chamber (add C0-Ct in the reference chamber to Ct in the clam chamber), 3) determine Z for each time interval, 4) select maximum Z to calculate FR (since the authors indicate that they think maximum filtration is the best metric of filtration rate), 5) calculate Z for the entire 4 hours as another way to compare between the two tidal flats. After this process has been done, then the authors can carry out t-tests on n=2 samples per site, for each of the components in the water, and for maximum and 4-hour filtration rates.

Table 2: Fresh weight is always a wet weight. So Fresh dry weight does not make sense. Are these values the Meat dry weights? Also on line 189.

Error bars on the figures: Thank you for specifying that the error bars represent 95% CI, but that does not address the whole issue I brought up last review. Is this the 95% CI based on n=2 (number of chambers), and if not, what is considered a sample?

Line 29: Rewrite – Clams were smaller at Sihwa than Geunso sites, but the filtration rate of 50 clams per chamber was higher at Sihwa than Geunso. This difference was probably not due to the immediate environmental conditions, since the enclosed chambers experienced no net current or tidal exchange during the 4-hour monitoring interval, and the initial seston and chlorophyll concentrations in the chambers were similar. Instead, these filtration rates may differ due to the hydrographic regime, since Sihwa tides are limited by sluice gates. Generally, at Geunso relative to Sihwa, current speeds are faster, and submergence times are shorter. Sihwa typically has higher chlorophyll a concentrations, as well as better food quality based on C/N ratio of POM (Sihwa 12.8, Geunso 15.6). The endogenous circatidal rhythm…

I am not necessarily convinced that the sentence about endogenous circatidal rhythm follows from the data, since there are no data showing that filtration rate changes over time more at Geunso than at Sihwa.

Line 39: There is no information about the effects of hydrographic regime, which implies that causality has been tested. This sentence should be rewritten: These findings suggest that hydrographic regime could be important in understanding in situ filtration rates.

These issues in the abstract also apply to the discussion.

Reviewer #2: The manuscript has been revised according to the comments. The questions have been answered well. The manuscript is recommended to be published in this journal.

7. PLOS authors have the option to publish the peer review history of their article (what does this mean?). If published, this will include your full peer review and any attached files.

Reviewer #1: Yes: Jennifer Ruesink

Reviewer #2: No

---

## [Author Response · Author response to Decision Letter 1]

7 Jan 2020

Reviewer #1: As I wrote before, it is not appropriate to combine measurements of three different components of the seston (POC, PON, chla), all collected in the same chamber at the same time, to calculate a mean filtration rate. Mean filtration rate could be based on the filtration of the same component of the water across the two experimental chambers at each site, after each is adjusted for the change in the companion reference chamber. A t-test could then be done using two samples at each site, since there were two experimental chambers (containing clams) at each site. (Note: I am actually uncertain whether the “four chambers” on line 155 refer to four chambers per tidal flat, or four chambers in total. If the latter, then there is not a possibility to compare the two tidal flats statistically, as there is only one filtration value for each flat. But in their response the authors write “two experimental chambers represent replicates” so I think they should be able to do statistics with n=2 at each tidal flat.)

→ The mean filtration rate (three different components of seston) was deleted according to your suggestion. The four chambers represent two experimental chambers (with clams) and two control chamber (without clams) at each site. Thus, we have done t-test using two samples at each site.

Line 230: How were the control chambers used in the calculations of filtration rate? The authors write: “the filtration rate was calculated for the period in which the maximum slope of the decreasing indicator concentration was observed.” There are three issues to address: 1) The maximum slope at Sihwa (Fig 5) occurs in the first time interval, but is equivalent for clam and reference chambers, such that no net filtration would be indicated IF the values were properly adjusted by comparison with reference; 2) the maximum slope seems to indicate the steepest portion of Figures 5 and 6, rather than the portion with the largest exponent in the equation for filtration; 3) selecting a time of maximum filtration from among the time periods creates potential bias if clam filtration at one site is more variable than at the other. The authors should do the following: 1) determine which reference chamber to pair with each clam chamber, 2) adjust Ct in the clam chamber so that it accounts for change in the reference chamber (add C0-Ct in the reference chamber to Ct in the clam chamber), 3) determine Z for each time interval, 4) select maximum Z to calculate FR (since the authors indicate that they think maximum filtration is the best metric of filtration rate), 5) calculate Z for the entire 4 hours as another way to compare between the two tidal flats. After this process has been done, then the authors can carry out t-tests on n=2 samples per site, for each of the components in the water, and for maximum and 4-hour filtration rates.

→ The filtration was recalculated according to your suggestion (Table 3). The experimental chambers (with clams) were paired up with each control chamber (C1-R1/C2-R2, Fig. 2) and then Ct value of experimental chambers was adjusted by each control chamber value (C0-Ct). The maximum Z among time periods was selected and then filtration rate was calculated. A t-test was carried out based on two samples at each site as well as each component (POC, PON and Chl a). The filtration rate during entire experimental period was described in Table S2.

Table 2: Fresh weight is always a wet weight. So Fresh dry weight does not make sense. Are these values the Meat dry weights? Also on line 189.

→ It was corrected to flesh dry weight in Table 2 and manuscript.

Error bars on the figures: Thank you for specifying that the error bars represent 95% CI, but that does not address the whole issue I brought up last review. Is this the 95% CI based on n=2 (number of chambers), and if not, what is considered a sample?

→ The statistical analysis was conducted based on two samples as we noted above.

Line 29: Rewrite – Clams were smaller at Sihwa than Geunso sites, but the filtration rate of 50 clams per chamber was higher at Sihwa than Geunso. This difference was probably not due to the immediate environmental conditions, since the enclosed chambers experienced no net current or tidal exchange during the 4-hour monitoring interval, and the initial seston and chlorophyll concentrations in the chambers were similar. Instead, these filtration rates may differ due to the hydrographic regime, since Sihwa tides are limited by sluice gates. Generally, at Geunso relative to Sihwa, current speeds are faster, and submergence times are shorter. Sihwa typically has higher chlorophyll a concentrations, as well as better food quality based on C/N ratio of POM (Sihwa 12.8, Geunso 15.6). The endogenous circatidal rhythm…

I am not necessarily convinced that the sentence about endogenous circatidal rhythm follows from the data, since there are no data showing that filtration rate changes over time more at Geunso than at Sihwa.

→ It was corrected as follows : The filtration rate of clams for POM at Sihwa tidal flat (2.86 for POC, 2.29 for PON and 5.46 L h-1 gDW-1 for Chl a) was higher than that at Geunso tidal flat (0.61 for POC, 0.89 for PON and 2.54 L h-1 gDW-1 for Chl a) which resulted from differences in the hydrographic regime, including tide characteristics, current speed and submergence time, and food quantity and quality (L-29). The description of endogenous circatidal rhythm was deleted in the manuscript.

Line 39: There is no information about the effects of hydrographic regime, which implies that causality has been tested. This sentence should be rewritten: These findings suggest that hydrographic regime could be important in understanding in situ filtration rates.

→ It was corrected according to your suggestion as follows : These findings suggest that hydrographic regime could be important in understanding in situ filtration rates of R. philippinarum.

---

## [Editor Report · Decision Letter 2]

16 Jan 2020

PONE-D-19-15055R2

Filtration rates of the manila clam, Ruditapes philippinarum, in tidal flats with different hydrographic regimes

PLOS ONE

Dear Dr Koo,

Thank you for submitting your manuscript to PLOS ONE. After careful consideration, we feel that it has merit but does not fully meet PLOS ONE’s publication criteria as it currently stands. Therefore, we invite you to submit a revised version of the manuscript that addresses the points raised during the review process.

ACADEMIC EDITOR: I am pleased with how the MS has taken shape for the better and would like to invite you to submit minor edits and clarifications as presented in the annotated pdf copy of the revised MS. 

We would appreciate receiving your revised manuscript by Mar 01 2020 11:59PM. To enhance the reproducibility of your results, we recommend that if applicable you deposit your laboratory protocols in protocols.io, where a protocol can be assigned its own identifier (DOI) such that it can be cited independently in the future. For instructions see: http://journals.plos.org/plosone/s/submission-guidelines#loc-laboratory-protocols

We look forward to receiving your revised manuscript.

Kind regards,

Ismael Aaron Kimirei, Ph.D.

Academic Editor

PLOS ONE

---

## [Author Response · Author response to Decision Letter 2]

18 Jan 2020

→ It was corrected as follows : Dissolved oxygen was monitored to prevent a decrease in metabolic activity due to depletion of dissolved oxygen in the chamber, and when the concentration decreased below 80%, oxygen was supplied using an aerator installed inside the chamber (L141-143).

→ 500 ml is correct volume. We did not use all water samples (750 ml except 250 ml for chlorophyll a analysis) for SS, POC and PON analysis (L 205).

→ It was corrected as follows : Thereafter, no significant changes in the chlorophyll a concentration in the Ruditapes chamber occurred, while that in the control chamber tended to increase (Table S1, L359-361).

→ It was corrected as follows : Han et al. (2008) reported that the filtration rate of small Ruditapes philippinarum (0.2 gDW) was about 28% higher than that of large ones (0.4 gDW) at 20 °C. This indicated that the filtration rate of Ruditapes can be changed by size and biomass, but, the difference in filtration rate is too great to be explained by size and biomass in this study (L 404-407)

→ It was corrected as follows : One experimental condition that must be considered carefully when calculating the amount of POM removed from seawater by filter feeders using a closed chamber, as indicated in this study, is that the concentration of POM introduced into the chamber at the beginning of the experiment should be maintained (L408-411).

→ It was corrected as follows : The relatively high filtration rate at Sihwa tidal flat can be attributed to this difference in hydrographic regimes. Another factor affecting the difference in filtration rate is high productivity in the water column at Sihwa tidal flat, which causes a quantitative difference in the food available to Manila clams. The Manila clams at Sihwa tidal flat had a higher growth rate than those at Geunso tidal flat, and the mean chlorophyll a concentration in the water column was significantly higher at the former site than the latter, with values of 3.1 and 52.5 μg L-1, respectively (L435-440) .

→ It was corrected as follows : The filtration rates obtained in this study, especially at Sihwa tidal flat, were higher than those reported in previous studies. While Lee [21] in similar field experimental settings as the current study reported the filtration rate of the Manila clams ranging from 0.13 to 0.97 L h-1 gDW-1, our study reports significantly higher values (2.29 to 5.46 L h-1 gDW-1), which could be related to differences in experiment methods. This value is significantly lower than the experiment results of the present study (L452-456).

→ It was corrected as follows : In another field experiment, Aoyama and Suzuki [23] reported the filtration rate of 3.44 L h-1 gDW-1 (L 464-465).

→ It was corrected as follows : In conclusion, this study evaluated differences in the filtration rate of R. philippinarum due to differing hydrographic regimes between two tidal flats that the influence of differing hydrographic conditions on the filtration rates of Manila clams has not been studied previously. Though this study was limited by factors such as salinity, turbidity and particle size that affect variations in filtration rates, the findings provide important information about the effects of hydrographic regime on the filtration rate of this species (L472-475).

→ We added this sentence : We would like to thank Dr. Jennifer Ruesink and anonymous reviewer for their valuable comments.

---

## [Editor Report · Decision Letter 3]

27 Jan 2020

Filtration rates of the manila clam, Ruditapes philippinarum, in tidal flats with different hydrographic regimes

PONE-D-19-15055R3

Dear Dr. Koo,

We are pleased to inform you that your manuscript has been judged scientifically suitable for publication and will be formally accepted for publication once it complies with all outstanding technical requirements.

With kind regards,

Ismael Aaron Kimirei, Ph.D.

Academic Editor

PLOS ONE

Additional Editor Comments (optional):

In the discussion, LN 456, please delete this sentence: "This value is significantly lower than the experiment results of the present study" Because it is already captured in the preceding sentence.
---

## [Editor Report · Acceptance letter]

30 Jan 2020

PONE-D-19-15055R3 

Filtration rates of the manila clam, *Ruditapes philippinarum*, in tidal flats with different hydrographic regimes 

Dear Dr. Koo:

I am pleased to inform you that your manuscript has been deemed suitable for publication in PLOS ONE. Congratulations! Your manuscript is now with our production department. 

With kind regards,

on behalf of

Dr. Ismael Aaron Kimirei 

Academic Editor

PLOS ONE